# Chromosomal Microarray in Patients with Non-Syndromic Autism Spectrum Disorders in the Clinical Routine of a Tertiary Hospital

**DOI:** 10.3390/genes14040820

**Published:** 2023-03-29

**Authors:** Ana Karen Sandoval-Talamantes, María Ángeles Mori, Fernando Santos-Simarro, Sixto García-Miñaur, Elena Mansilla, Jair Antonio Tenorio, Carolina Peña, Carmen Adan, María Fernández-Elvira, Inmaculada Rueda, Pablo Lapunzina, Julián Nevado

**Affiliations:** 1INGEMM, Institute of Medical and Molecular Genetics, La Paz University Hospital, IdiPAZ, 28046 Madrid, Spain; 2ITHACA, European Research Network, La Paz University Hospital, 28046 Madrid, Spain; 3Network for Biomedical Research on Rare Diseases (CIBERER), Carlos III Health Institute (ISCIII), 28046 Madrid, Spain

**Keywords:** autistic spectrum disorder, copy number variations, microarray, tertiary hospital

## Abstract

Autism spectrum disorders (ASD) comprise a group of neurodevelopmental disorders (NDD) characterized by deficits in communication and social interaction, as well as repetitive and restrictive behaviors, etc. The genetic implications of ASD have been widely documented, and numerous genes have been associated with it. The use of chromosomal microarray analysis (CMA) has proven to be a rapid and effective method for detecting both small and large deletions and duplications associated with ASD. In this article, we present the implementation of CMA as a first-tier test in our clinical laboratory for patients with primary ASD over a prospective period of four years. The cohort was composed of 212 individuals over 3 years of age, who met DSM-5 diagnostic criteria for ASD. The use of a customized array-CGH (comparative genomic hybridization) design (KaryoArray^®^) found 99 individuals (45.20%) with copy number variants (CNVs); 34 of them carried deletions (34.34%) and 65 duplications (65.65%). A total of 28 of 212 patients had pathogenic or likely pathogenic CNVs, representing approximately 13% of the cohort. In turn, 28 out of 212 (approximately 12%) had variants of uncertain clinical significance (VUS). Our findings involve clinically significant CNVs, known to cause ASD (syndromic and non-syndromic), and other CNVs previously related to other comorbidities such as epilepsy or intellectual disability (ID). Lastly, we observed new rearrangements that will enhance the information available and the collection of genes associated with this disorder. Our data also highlight that CMA could be very useful in diagnosing patients with essential/primary autism, and demonstrate the existence of substantial genetic and clinical heterogeneity in non-syndromic ASD individuals, underscoring the continued challenge for genetic laboratories in terms of its molecular diagnosis.

## 1. Introduction

Autism spectrum disorders (ASD) are neurodevelopmental alterations characterized by deficits in communication and social interaction, as well as repetitive and restricted behaviors, anxiety, hyperactivity, or experienced arousal [1,2]. It includes typical childhood autism and other clinical phenotypes (such as Asperger syndrome) and generalized developmental disorders not otherwise specified in other categories [2]. According to the International Classification of Diseases (ICD-11.1) and the fifth edition of the Diagnostic and Statistical Manual of Mental Disorders (DSM-5), ASDs are categorized as generalized developmental disorders. ASD can be non-syndromic (primary, essential) or syndromic (related to more than 100 different genetic disorders) [3]. In terms of health, ASD is the most common neuropsychiatric disorder, beginning in the early years of life and it affects 1.43 per 1000 people [4]. The recurrence risk is generally considered 5–10% (7% for females and 14% for males, [5]). Thus, ASD is most common in males, with a male/female ratio of 4:1 [4]. In Spain, the prevalence of ASD in the pediatric population ranges from 0.64% to 2% [6,7,8,9]. ASD is considered to be a genetic-based disease. There is a concordance of over 88% for monozygotic and over 30% for dizygotic twins [10], suggesting a multifactorial genetic condition. The discovery of genetic involvement in the etiology of ASD has made this condition a strong candidate for genomic-based diagnostic tests, including chromosome microarrays (CMA) and next-generation sequencing (NGS). The identification of copy number variants (CNVs) using single-nucleotide polymorphism (SNP) or comparative genomic hybridization (CGH) microarrays has proven to be a rapid and effective method for detecting both small and large deletions and duplications associated with ASD [11]. Since 2010, the International Standards for Cytogenomic Arrays (ISCA) Consortium and other institutions, such as the Spanish Human Genetic Association (AEGH), as well as different publications [12,13], have recommended the use of CMA as a first-tier test for patients with intellectual disabilities (ID), congenital malformations and ASD.

A recent study using a genomic approach by CMA in individuals with ASD revealed that CNVs occurred in 10 to 20% of cases [14]. Thus, it has been estimated that there are 130 to 234 ASD-related CNV regions in the human genome [15]. Whole genome association studies (GWAS) have also established more than 20 genetic loci with a high risk of susceptibility to autism. After mapping at chromosomes 1p, 2q, 5q, 7q, 15q, 17q 19p, and Xq [15,16], the most frequently reported CNV is 16p11.2 deletion (approximately 1% of cases) followed by 7q11.23, 17p11.2, and 22q11 duplications [17]. Other regions involved include duplications in 15q11.2, or Xp22.33/Yp11.32-p11.31 [18]. Regarding the use of CMA in Spain, few studies have been conducted on the interpretation of CNVs in Spanish ASD patients, which have included syndromic and non-syndromic ASD individuals with other comorbidities [18,19,20,21,22,23,24]. Most studies focused on individuals with ASD, ID, congenital malformations, or other neurodevelopmental alterations. Therefore, few chromosomal regions have been identified for ASD individuals, with results ranging from 2.6% to 12.5% (average 8.65%) in Spain [18,19,20,21,22,23,24]. CMA analysis has been widely used in ASD studies in many other countries, such as Canada, the United States, China, Saudi Arabia, Finland, Taiwan, Tunisia, Portugal, Italy, etc., with clearly significant results ranging from 2.7 to 22.5% [3,4,11,15,25,26,27,28,29,30,31,32,33,34]. 

This study uses CMA as a first-line test in patients with primary ASD in a tertiary hospital during a prospective period of four years. The aim is to assess the putative genes or regions involved and evaluate the efficiency of CMA as a first-tier test for individuals with non-syndromic ASD.

## 2. Materials and Methods

### Subjects 

Sample recruitment was carried out prospectively, from 2016 to 2019 mostly, among patients under routine care in the genetic or neuropediatric clinics of La Paz University Hospital in Madrid, Spain (a tertiary-level hospital). Patients with a previously abnormal karyotype in peripheral blood, a positive Fragile-X syndrome study, or a confirmed suspicion of an identifiable syndromic entity were excluded from the study (see Figure 1). In the beginning, patients who underwent diagnostic evaluation by geneticists or neurologists that showed a genetically validated and identifiable monogenic disease or syndrome, multifactorial cause, or exposure to teratogens were excluded as well (Figure 1). Each individual had a genetic examination and clinical evaluation of their cognitive abilities and ASD symptoms, to ensure that only subjects with non-syndromic ASD phenotypes were included. After this evaluation, 232 patients aged >3 years with apparently non-syndromic ASD and who met DSM-5 diagnostic criteria were initially recruited for the study. In the end, other cases were excluded if the DNA quality prevented the CMA assay's performance and there was no possibility of collecting a new sample. The final cohort was composed of 212 individuals. Data collection and sampling were conducted with prior written informed consent from the subject. 

## 3. Genetic Studies

### 3.1. Karyotyping and FISH Analyses

Cytogenetic analyses were performed on GTG-banded metaphases at a resolution of approximately 550 bands. Analyses were carried out according to standard laboratory protocol using Chromosome Kit P (Euroclone; Siziano, PV, Italy). Fluorescence in situ hybridization (FISH) analyses were performed according to standard laboratory protocols using different probes (Kreatech Biotechnology B.V, Amsterdam, The Netherlands, and Vysis Inc.; Downers Grove, IL, USA).

### 3.2. Chromosomal Microarray Analysis (CMA) 

CMA analysis was performed by comparative genomic hybridization (aCGH) in all 212 patients using our custom platform (KaryoArray^®^v3.0, [35,36]). The microarray design used for the analysis was SurePrint G3 CGH 8 × 60 K (Agilent Technologies; Santa Clara, CA, USA). The design focused on more than 375 regions of high clinical significance in the human genome. It had a higher resolution than the ISCA 8 × 60 K model (comparable to a 240 K array), which was defined based on the ISCA Consortium and our own clinical experience [35,36]. Data were analyzed using CytoGenomics software (Agilent Technologies) with the default CGH analysis method and interpreted with reference to available databases (DGV, ClinVar, DECIPHER, STRING, etc.). Classification and interpretation of each of the variants was categorized as benign/likely benign (B/LB), pathogenic/likely pathogenic (P/LP), or of uncertain clinical significance (VUS), according to the American College of Medical Genetics (ACMG/AMP, see [37]).

### 3.3. MLPA (Multiplex Ligation-Dependent Probe Amplification)

Several MLPA Salsa kits were used to confirm some of the CNVs found. To rule out subtelomeric rearrangements, we applied MLPA kits P036 and P070, along with Salsa kits (including Salsas P245, P373, P064), for several microdeletion/microduplication syndromes (MRC-Holland; Amsterdam, The Netherlands). Data analyses were performed according to the manufacturer’s protocols. The relative probe signals were defined by dividing each measured peak area by the sum of all peak areas of the control probes of that sample. The ratio of the relative probe area of each peak was then compared to that obtained on a DNA control sample using Coffalysser v.9.4; (MRC-Holland; Amsterdam, The Netherlands).

### 3.4. Fragile-X Syndrome Analysis

Repeated CGG in the FXS region of the FMR1 gene involved was quantified by TP-PCR, using a commercial kit following the manufacturer’s instructions (LabGscan FRAXA; Diagnostica LongWood; Zaragoza, Spain). The products were migrated into the 3130xl ABIPrism genetic autoanalyzer (Thermo-Fischer, Waltham, MA, USA).

### 3.5. Statistical Analysis

Statistical analysis was performed with SPSS version 25 (IBM Corporation; Armonk, NY, USA). The two groups were compared by Student’s *t*-test for continuous variables and chi-square tests for categorical variables. Comparisons between several groups were carried out by ANOVA analysis and Bonferroni post hoc tests for continuous variables and z-tests of column proportions for categorical variables. A *p*-value (observed significance level) of less than 0.05 was considered to indicate a statistically significant difference.

### 3.6. Limitation of the Study

One of the difficulties in carrying out this study was finding individuals with primary autism who did not show malformations or any kind of dysmorphia when examined by clinical experts. The limited sample size and lack of functional/segregation analysis in uncertain CNVs may also impact this study.

## 4. Results

The cohort was composed of 212 individuals with a mean age of 10.73 ± 6.42 years and a median age of 10. The male/female ratio was 4.04:1 (186/46 cases, respectively), with no statistically significant differences between mean ages by gender, which were 10.58 ± 6.13 years for males and 11.20 ± 6.86 years for females (Student’s-*t*-test: *p* = 0.571). The patients in the cohort were mostly of pediatric age (<16 years, 189 individuals, 89.15%), and all were from Spain. Although the primary reason for visiting our clinic was autism (173/212, 81%), we also identified 24 individuals with Asperger’s syndrome (11.32%), 17 (8%) with a diagnosis of attention deficit and hyperactivity disorder (ADHD), and 15 with non-specific developmental disorders (7.7%). In addition to ASD, we also detected the presence of other comorbidities, with intellectual disability and psychomotor delay being observed most frequently (69 patients, 32%). In addition, 6.6% (14 individuals) were diagnosed with epilepsy but in none of them could any syndromic entity can be recognized.

Using KaryoArray^®^ [35,36] as a first-tier test in patients with primary ASD, we found B/LB, P/LP, and VUS CNVs in 99 out of 212 individuals (46.70% of the cases). No CNVs were observed in the remaining 113 (53.32%) individuals. We found 130 CNVs in the 99 patients showing rearrangements (average, 1.28 CNV/patient; median size, 158 kb; mean, 802.83 ± 1952.23 kb). Where 82/99 individuals carried a single CNV, 10/99 patients carried two CNVs, 5/99 carried three and only 2/99 carried four. Thus, multiple CNVs were found in 17 patients (17.17% of the total with CNVs). Interestingly, patients with pathogenic CNVs carried single rearrangements, and patients with B/LB or VUS variants carried several of them. In addition, individuals with deletions seem to be less represented than those with duplications, at a ratio of 1:1.90 (58/99; 58.59% versus 29/99; 29.29%, respectively). A total of 12/99 individuals showed both deletions and duplications (12.12%). This ratio (del/dup) increased slightly when the analysis was limited to individuals with pathogenic and likely pathogenic cases, finding 32/52 (61.50%) with duplications versus 11/52 (21.15%) with deletions, at a ratio of 1:2.91. When analyzing by CNV type rather than by individuals, duplications were also more frequent (n = 80, 62.20%; median size, 218.30 Kb) than deletions (n = 50, 37.80%; median size, 80.67 Kb). Representative cases of large and small CNVs (duplications/deletions) are shown in Figure 2. 

In terms of individuals, 28 of 212 patients carried P/LP CNVs, representing 13.2% of the sample; 45/212 showed benign rearrangements, while 26 of 212 had VUS CNVs (see Figure 3). 

The familial segregation of CNVs was only studied in 11 of 28 P/LP patients. We found five de novo pathogenic variants, four of maternal origin and two of paternal origin. Table 1 and Table 2 summarize the findings and classification of the P/LP and VUS variants using ACMG/AMP criteria [37].

Interestingly, B/LB CNVs were smaller in size (median, 66.95 Kb; mean, 142.06 ± 215.81) than both VUS (median, 490 Kb; mean 585.72 ± 836.01) and P/LP (median, 742.67 Kb; mean, 2202.37 ± 3300.41). Statistically significant differences were observed among these three categories (one-way ANOVA *p* = 0.0001, F = 16,130, post hoc, T3-Dunnett, between B/LB and VUS, B and LP/P, VUS and LP/P) (Table 3). Table 3 also summarizes statistically significant size differences between B/LB and LP/P (*p* < 0.05, Student’s *t*-test) at the three ranges of CNV sizes analyzed (<0.1; 0.1–1 Mb and >1 Mb), regardless of the type of rearrangement (deletions or duplications). VUS CNVs had an intermediate size between B/LB and P/LP variants. 

## 5. Discussion

Autism spectrum disorders (ASDs) constitute a group of severe neurodevelopmental conditions characterized by an increase in the “informal” inclusion of different aspects (anxiety, experienced arousal [1]; early motor signs [38]) with complex multifactorial etiology. It is widely accepted that evaluations of patients with autism/ASD may involve a CMA as a first-tier test under different recommendations [12,13]. Nevertheless, the situation in Spain during the enrollment period for this study was different. We realized that CMA was not frequently used in our country, at least not in most public hospitals. In 2017, only 16.28% of the network of public hospitals in Spain fully addressed these recommendations (Spanish Human Genetics Association, AEGH; unpublished data). This fact could compromise, in a significant part, the diagnostic yield in ASD individuals, among others. Genetic studies for individuals with ASD may also be compromised by the fact that ASD is not diagnosed through biomarkers but through a process of observation and a systematic cognitive/behavioral examination. In addition, the irruption of other genomic technologies, such as NGS, competing as the first line of testing in patients with ASD [19,20,22,23] can significantly delay the entire diagnostic process, for economic reasons, at least in non-tertiary hospitals in our country. Ironically, none of these patients have a simple genetic study, as they are waiting for a trio of Exome analyses. Therefore, since, for us, the clinical and genetic diagnosis of individuals with ASD is critical, we encourage our colleagues to fight against this circumstance, namely the insufficient use of new genomic technologies such as CMA and NGS (as a first test or combined), in the routine clinical diagnosis of ASD. Data within this work support that CMA could also be very useful to segregate cases of individuals with apparently essential/primary autism.

Presently, only a few CNVs are reported in databases as being associated with essential ASD clinical features (OMIM, https://www.omim.org/, accessed on 27 February 2023), and most of them are characterized by incomplete penetrance and highly variable expressivity. Few studies have been conducted on the interpretation of CNVs in Spanish ASD patients, most of which have been extracted from a heterogeneous group of patients with global neurodevelopmental delay [18,19,20,21,22,23,24]. In this study, we used a previously validated and trained CMA platform [35,36] for the screening of apparently (under clinical geneticists’ supervision) non-syndromic ASD individuals, which allowed us to segregate up to 87% of individuals in the cohort or to classify CNVs in approximately 75% of the patients showing CNVs (see Figure 3). These rates are similar to or even higher than those previously published for global ASD in our country or nearby [19,23,24,30,32,34]. Regarding the quick segregation mentioned before, we were able to establish a statistically significant relationship between the size of the CNVs and their clinical classification (under ACM/AMP criteria) in primary ASD patients. In fact, according to previous work, P/LP CNVs are also larger (including more genes) than B/LB variants (Monteiro, Pinto) in essential cases of ASD. Interestingly, VUS had an intermediate size between B/LB and P/LP CNVs. Like other previous studies, CNV gains were also more frequent than deletions (Monteiro, Pinto). Our results also supported the theory that a well-known platform by the user, the selection of the patients, and the use of their databases may help improve CMA results in general, also including individuals with primary ASD.

We found 28 cases carrying P/LP CNVs associated with apparently primary ASD (approximately 13% of the subjects) that were similar to previous reports in global ASD individuals [20,30,34] in the literature and slightly higher than other studies reported in our country [19,22,23], but is difficult to compare because many of the studies do not segregate between isolated and non-isolated ASD subjects. We also observed VUS variants in 26 of 212 cases (12.26%). This number is higher than other studies addressing VUS in ASD individuals in Spain [20,22] or our previous report for this platform in patients with ID, congenital malformations, or ASD, together (estimated in 3%, during its validation [35] or up to 7% within its clinical implementation process; unpublished data). This platform covers clinically relevant regions of genomic imbalances maintaining an acceptable balance between the number of VUS and known/unknown genomic aberrations [35]. Thus, this fact may give us stronger support for these regions, CNVs, or genes to be putatively associated with primary ASD, as the platform used combines genome-wide coverage with targeted clinically validated P/LP enrichment regions. 

Overall, CNVs (P/LP or VUS) found may involve regions that play a role in several neurodevelopmental disorders and probably contribute substantially to either ASD or ASD risk/susceptibility. Among them, we highlight some candidate regions that have not been strongly associated with ASD such as (i) duplications at: Xp22.33, 2p25.3, 3p26.3, 3p26.3-p26.2, 6q16.3, 9p24.3, 9q34.13, 15q11.2, 22q11.21 (ii) deletions at: Xp22.33-p22.2, 2q23.1, 3p25.3-p25.2, 5p15.33, 8q21, 9p24.3, 10q21.3, 15q26.1 y 18q21. Interestingly, although the primary focus for selecting these cases was non-syndromic ASD, an important number of affected regions and genes found are related to syndromes, ID, and other comorbidities. They can be grossly divided into four groups:

The first group includes genes that are already known to be related to ASD. An example is individual AUT45, who presented a gain in the *ZNF44* gene, a gene previously associated with ASD (https://gene.sfari.org/ accessed on 3 November 2022 and [39,40]). This patient also had a duplication in the *CNTN4* gene. Deletions and duplications of the *CNTN4* gene, which plays an important role in neuronal maintenance and plasticity, have also been described in autism patients [41]. Individual AUT164 presented a deletion including the *SLC6A1* gene, which has an established role in ASD [42]. We also found one patient (AUT28) with a duplication in the Xp22.33 region, associated with autism. This characteristic has been already reported in the DECIPHER database in nine patients, (#285674, 284752, 290397, 325813, 351449, 314771, 328499, 289827, and 333223). Another example is the duplication involving the *CHL1* gene (3p26.3), also reported associated with ASD [43,44,45]. One of the genes frequently observed in copy number variations in this cohort was *CHRNA7*. The role of deletions and duplications of *CHRNA7* remains controversial, but they have also been associated with ASD in several research articles, and includes single-nucleotide variants (SNVs) within this gene. *CHRNA7* deletions are considered pathogenic (AUT230), although it is difficult to establish whether the relationship is causal or involves susceptibility to ASD. On the other hand, duplications in this gene are classified as VUS (patients AUT224, AUT225, AUT227, AUT229, AUT231), because *CHRNA7* duplications were observed both in individuals with ASD and in controls (apparently without ASD) [46].

A second group of CNVs involved a significant number of syndromic deletions/gains (12/28 cases; 40.85%) that are already known to be related to ASD [47,48,49,50,51,52,53,54,55,56,57,58,59,60]. They include 16p11.2-p12.2 duplication (reciprocal to OMIM#613604, two patients,), 15q11.2 duplication (including autism susceptibility in region 4, part of OMIM#608636, two patients), 22q13 deletion or Phelan–McDermid syndrome (OMIM#606232, two patients), 2q11.1-q13 duplication (one patient), 2q37 deletion (OMIM#600430, one patient,), 3q29 deletion (OMIM #611936, one patient), 1q21.1 duplication (OMIM #612475, one patient), including only BP3 and a small part of BP4. We also observed that duplications in 15q11.2-q13 (OMIM #608636) seemed to occur in approximately 3% of this cohort, and are known to be maternally inherited (two patients, AUT97 and AUT136). In addition, we found a patient (AUT146) with an intronic duplication affecting *GABRG3*. However, we consider it to be a VUS because it is a 30 Kb deletion near exon-2, and we cannot rule out the possibility of involving additional probes using a higher-resolution platform. The *GABRG3* gene in the 15q12 region has been suggested as a possible candidate to explain ASD in several patients [54]. Regarding other syndromic regions, we highlighted a deletion in 18q21.2 encompassing the *TCF4* gene (patient AUT36). Defects in this gene are a cause of Pitt–Hopkins syndrome (OMIM #610954), which seems to correspond with certain clinical features of the findings in this patient after re-evaluating. The deletion in 15q26 (AUT81), a region belonging to the 15q26-qter microdeletion syndrome, had not been previously reported in ASD patients, but it has been associated with ID [55]. However, this region includes *CHD2*, a gene related to ASD in recent works and some databases (Simons Simplex Collection and SFARI database, with evidence in at least ten articles). This gene is also involved in the development of the central nervous system and belongs to the immunoglobulin superfamily of cell adhesion molecules [45,56]. We also remark additionally a duplication in 9q34.13 (AUT95), which affects the *POMT1* gene. This gene has been associated with Walker–Warburg syndrome (OMIM #236670), C1-girdle muscular dystrophy (OMIM #609308), and intellectually impaired muscular dystrophy (OMIM #613155). *POMT1* has also been associated with ASD in the SFARI database, a dosage-sensitive gene that appears to contribute to the pathogenesis of primary ASD [57]. We also observed a duplication in 22q11.2 (AUT185) associated with the 22q11.2 microduplication syndrome (OMIM #608363). Patients with ASD have been reported to be associated with duplication in 22q11.2 [58,59]. This region includes the *SEPT5*, *GP1BB*, and *TBX1* genes. Finally, another interesting example was a deletion in 5p15.33, located in a region that includes 5p- or Cri du Chat syndrome (OMIM# 123450) and is associated with psychiatric behavior (auto-aggression, hyperactivity deficit, sleep disturbances, etc.) [60]. However, this particular patient also had a duplication in the 2p25.3 region, suggesting a derived chromosome, as a result of a balanced translocation in the father. Detecting balanced translocations in parents has important implications for family genetic counseling.

The third group of CNVs includes some of the regions that not only correspond to ASD traits but which may also be related to different comorbidities including psychiatric disorders, aggressive behavior, mood swings, epilepsy, or ID. This is the case of abnormalities in the *DOCK8* gene located in 9p24.3, which have also been associated with type-2 mental retardation [61]. We identified two patients affected with pathogenic variations in this region, one carrying a deletion (AUT46, who presented autism) and one with a duplication (AUT194). Several articles correlate deletions and duplications in this region with ASD [62]. In addition, *DOCK8* is included in the list of genes associated with ASD in the AutDB database [63]. In the Autism Sequencing Consortium cohort, there are patients with de novo loss of function variants associated with the *DOCK8* gene in patients with ASD. Patient AUT35 carried a duplication in the *GRIK2* gene region at 6q16.3, which has also been described as associated with autism (DECIPHER#284729). Although this gene has been mainly related to ID and epilepsy [64], there is a literature report of another patient with a deletion in the gene who was also associated with autism [65]. In addition, an *MBD5* deletion was found in patient AUT218. *MBD5* is a member of the methyl-CpG-binding domain (MBD) family of proteins. Haploinsufficiency of this gene is associated with a syndrome involving microcephaly, ID, severe speech impairment, seizures, short attention span, autistic-like behaviors (gaze avoidance, inattention, and repetitive behaviors), and other stereotypic and repetitive behaviors (e.g., teeth grinding, hand chewing and repetitive hand movements) [66].

The fourth group of CNVs involves chromosomal regions with somewhat more variable phenotypes. For example, some of the clinical characteristics of patient AUT167 (with short stature, ASD, and ID) could be explained by the deletion found in Xp22.33-p22.2. This region includes the Xp22.3 microdeletion syndrome (OMIM # # 300830), which presents a highly variable phenotype depending on the size of the deletion. Nonetheless, it is primarily characterized by X-linked ichthyosis, mild to moderate ID, Kallmann syndrome, short stature, chondrodysplasia punctata, and ocular albinism. Epilepsy, attention deficit hyperactivity disorder, autism, and social communication disorder may also be associated. In patient AUT167, the deleted gene that seems to be associated with autism and ID is *NLGN4X* (whose protein participates in synaptogenesis and glutamatergic pathways), located in the region of susceptibility for autism and Asperger (OMIM #300497 and OMIM #300495) [67]. In other subgroups of patients, more than one CNV (two deletions) were found to be responsible for the phenotype (for example, patient AUT217). We observed a gene deletion of *IMPA1* in one of them and *CTNNA3* in the other. Disorders associated with *IMPA1* include ID, mental retardation, autosomal recessive 59, and bipolar disorder (https://www.omim.org, accessed on 3 November 2022). Mutations in *CTNNA3* are associated with familial arrhythmogenic right ventricular dysplasia-13, which is not ASD-related (https://www.omim.org, accessed on 3 November 2022).

Regarding uncertain CNVs, VUS are a critical aspect in the use of genomic technologies in clinical routine, as well as for appropriate genetic counseling. In the absence of more information, we believe that VUS should not be used to condition the clinical management of these patients, but it may potentially suggest new regions associated with ASD, from a research point of view. These variants will be reclassified as benign or pathogenic in the future, as more information becomes available. Thus, it is necessary to conduct a review of these variants in these particular patients with a consistent frequency. Among the VUS found in our cohort, we noted some examples of CNVs associated with ASD. One patient (AUT175) was found with a duplication in 17p11.2-p11.1 involving the genes *MAP2K3*, *KCNJ12*, *KCNJ18*, *C17orf51*, *FAM27L*, and *FLJ36000*. Recently, another patient with autism and ID showing a similar CNV was reported in the DECIPHER database (#24937). Patients AUT103 and AUT104 had a deletion in 8p11.21.1, affecting the *POMK* (*SGK196*) gene. Loss of function compound heterozygous variants in this gene were found in a patient affected by the Walker–Warburg syndrome (WWS) phenotype. We also found some VUS not previously associated with ASD, such as 1p32.3 duplication (AUT100), 1q44 duplication (AUT75), 3p25.3-p25.2 deletion (AUT169), 10q24.32 duplication (AUT129), 12q24.33 duplication (present in two brothers, both with ASD; AUT179 and AUT180), 14q13.2 duplication (AUT62), and 14q24.3 duplication (AUT148).

Finally, the present study has several limitations. One of the difficulties in carrying out this study was finding individuals with primary autism who did not show malformations or any kind of dysmorphia when examined by clinical experts. In addition, given the limited sample size and lack of parental segregation in many of these CNVs, we cannot define whether CNVs are de novo or inherited, at this point. This limitation is especially relevant for VUS variants. Furthermore, this study is focused on a customized aCGH of 60K. The use of a higher-resolution SNP array (850 K or higher) could give us other experimental points of view. 

## 6. Conclusions

Overall, these data may also support the idea that the etiology of the essential ASD may involve the functional alteration of inter-related genes in neuronal networks (see graphical abstract and [4,33]), and demonstrate substantial genetic and clinical heterogeneity in ASD individuals, underscoring the continued challenges in their molecular diagnosis. These findings also demonstrate that CMA could be very useful for the segregation of cases of individuals with apparently essential/primary autism. We established that approximately over 85% of the individuals in our cohort have been rapidly screened using CMA analysis, a cost-effective first-tier test in most laboratories in the world. This has allowed us to establish clinically relevant CNVs (around 13%) with important implications for genetic counseling for these families. Among them, we mainly found a significant number of recurrent variations [68]. Lastly, we documented several CNVs that have not been previously reported in ASD that may also influence the development of susceptibility to ASD, which needs further experimental approaches (such as NGS) and/or larger cohorts to confirm their role. 

## Figures and Tables

**Figure 1 genes-14-00820-f001:**
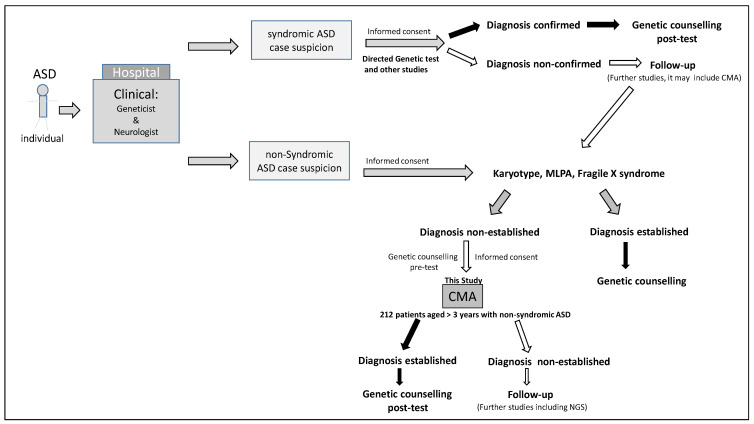
Algorithm used in patient selection. ASD—Autistic Spectrum Disorder); CMA—Chromosomal Microarray Analysis).

**Figure 2 genes-14-00820-f002:**
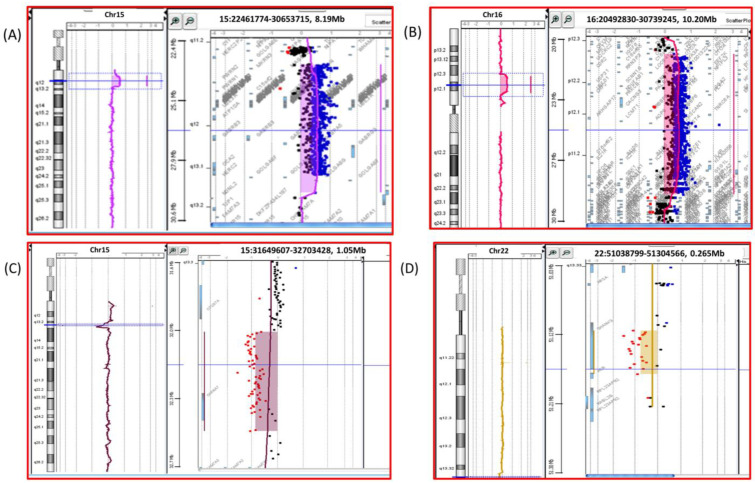
Representative cases of large and small duplications/deletions. (**A**) CMA shows an 8.19 Mb duplication on chromosome 15, (**B**) CMA shows a 10.2 Mb duplication on chromosome 16, (**C**) CMA shows a 1.05 Mb deletion on chromosome 15, and (**D**) CMA shows a 265 Kb deletion on chromosome 22.

**Figure 3 genes-14-00820-f003:**
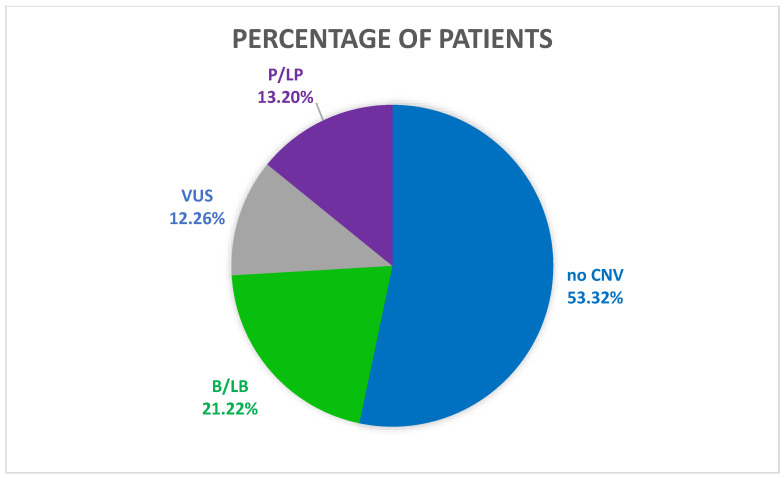
Percentage of individuals with P/LP, B/LB, and those with no CNVs found. Out of the 212 patients, 28 carried P/LP CNVs; 45/212 carried benign CNVs, and 26/212 carried VUS, while 113 (53.32%) did not show any CNVs.

**Table 1 genes-14-00820-t001:** Pathogenic and likely pathogenic variants found in the cohort of primary ASD individuals.

Patient	Clinical Features	Results	Size (Kb)	Genes. OMIM Genes Underlined	Classification
AUT24	ASD, seizures, and plagiocephaly	arr[hg19] 16p12.2p11.2(21475060_29182196)x3	8000	*OTOA*, *UQCRC2*, *SCNN1G*, *SCNN1B*, *COG7*, *EARS2*, *PALB2*, *TNRC6A*, *IL4R*, *IL21R*, *KIAA0556*, *CLN3*, *TUFM*, *ATP2A1*, *CD19*, *LAT*	P
AUT28	ASD, developmental delay and joint laxity	arr[hg19] chrXp22.33(194896_2658336)x3	2400	*PLCXD1*, *GTPBP6*, *NCRNA00107*, *PPP2R3B*, *SHOX*,*CRLF2*, *CSF2RA*, *IL3RA*, *SLC25A6*, *NCRNA00105*, *ASMTL*, *P2RY8*, *SFRS17A*, *ASMT*, *DHRSX*, *ZBED1*, *CD99, XG, GYG2, ARSD, ARSE*	LP
AUT33	ASD, developmental delay, seizures, and joint laxity	arr[hg19] chr3p26.3(297696_927607)x3	629.9	*CHL1*	LP
AUT35	ASD and hyperactivity	arr[hg19] 6q16.3(101869595_102582366)x3	712.7	* GRIK2 *	LP
AUT36	ASD, microcephaly, developmental delay, hypotonia, and seizures	arr[hg19] 18q21.2(51091726_51331417)x1	240	* TCF4 *	P
AUT45	ASD Asperger	arr[hg19] 3p26.3-p26.2(2163261_2876491)x3, mat	713	*CNTN4*	LP
		arr [hg19] 19p13.2(12388254_12403612)x3	15.3	*ZNF44*	LB
AUT46	ASD	arr[hg19] 9p24.3(300306_371857)x1	72	* DOCK8 *	LP
AUT49	ASD, developmental delay, ectopic kidney, hyperactivity, dolichocephaly, prominent forehead, hypoplasia of the middle third of the face, and short palpebral fissures	arr[hg19] 2q37.2q37.3(236959860_242852625)x1	5892	*AGAP1*, *GBX2*, *ASB18*, *IQCA1*, *CXCR7*, *COPS8*, *COL6A3*, *MLPH*, *PRLH*, *RAB17*, *LRRFIP1*, *RBM44*, *RAMP1*, *UBE2F*, *UBE2F*, *SCLY*, *ESPNL*, *KLHL30*, *ILKAP*, *LOC151174*, *LOC643387*, *HES6*, *PER2*, *TRAF3IP1*, *ASB1*, *LOC151171*, *TWIST2*, *FLJ43879*, *HDAC4*, *MGC16025*, *MIR4269*, *LOC150935*, *NDUFA10*, *OR6B2*, *PRR21*, *OR6B3*, *MYEOV2*, *OTOS*, *GPC1*, *PP14571*, *MIR149*, *ANKMY1*, *DUSP28*, *RNPEPL1*, *CAPN10*, *GPR35*, *AQP12B*, *AQP12A*, *KIF1A*, *AGXT*, *C2orf54*, *LOC200772*, *SNED1*, *MTERFD2*, *PASK*, *PPP1R7*, *ANO7*, *HDLBP*, *SEPT2*, *FARP2*, *STK25*, *BOKAS1*, *BOK*, *THAP4*, *ATG4B*, *DTYMK*, *ING5*, *D2HGDH*, *GAL3ST2*, *NEU4*, *PDCD1*, *C2orf85*	P
AUT52	ASD, developmental delay, hypotonia, and hearing loss	arr[hg19] 2p25.3(39193_4234087)x3	4190	*FAM110C*, *SH3YL1*, *ACP1*, *FAM150B*, *TMEM18*, *C2orf90*, *SNTG2*, *TPO*, *PXDN*, *MYT1L*, *LOC730811*, *TSSC1*, *TTC15*, *ADI1*, *RNASEH1*, *RPS7*, *COLEC11*, *ALLC*	P
		arr[hg19] 5p15.33(204849_3876457)x1	3670	*CCDC127*, *SDHA*, *PDCD6*, *AHRR*, *LOC100310782*, *C5orf55*, *EXOC3*, *LOC25845*, *SLC9A3*, *CEP72*, *TPPP*, *ZDHHC11*, *BRD9*, *TRIP13*, *NKD2*, *SLC12A7*, *SLC6A19*, *SLC6A18*, *TERT*, *CLPTM1L*, *SLC6A3*, *LPCAT1*, *SDHAP3*, *LOC728613*, *MRPL36*, *NDUFS6*, *IRX4*, *IRX2*, *C5orf38*, *IRX1*	P
		arr[hg19] 15q11.2(25420800_25491412)x3	70.61	*SNORD115-4*, *SNORD115-5*, *SNORD115-10*, *SNORD115-9*, *SNORD115-12*, *SNORD115-6*, *SNORD115-7*, *SNORD115-8*, *SNORD115-36*, *SNORD115-43*, *SNORD115-29*, *SNORD115-11*, *SNORD115-13*, *SNORD115-14*, *SNORD115-16*, *SNORD115-18*, *SNORD115-19*, *SNORD115-17*, *SNORD115-21*, *SNORD115-20*, *SNORD115-15*, *SNORD115-22*, *PAR4*, *SNORD115-23*, *HBII-52-24*, *SNORD115-25*, *SNORD115-26*, *HBII-52-27*, *HBII-52-28*, *SNORD115-30*, *SNORD115-31*, *SNORD115-32*, *SNORD115-33*, *SNORD115-34*, *SNORD115-35*, *SNORD115-37*, *SNORD115-38*, *SNORD115-39*, *SNORD115-40*, *SNORD115-41.*	B
AUT59	ASD, seizures, absence of speech, psychomotor delay	arr [hg19] 16p11.2(28846555_31713018)x3	2800	*ATXN2L*, *TUFM*,*SH2B*, *ATP2A1*, *CD19*, *NFATC2IP*, *SPNS1*, *LAT*, *RRN3P2*, *RUNDC2C*, *LOC606724*, *BOLA2*, *BOLA2B*, *GIYD2*, *GIYD1*, *SULT1A4*, *SULT1A3*, *LOC388242*, *LOC613038*, *LOC440354*, *SLC7A5P1*, *SPN*, *QPRT*, *C16orf54*, *ZG16*, *KIF22*, *MAZ*, *PRRT2*, *C16orf53*, *MVP*, *CDIPT*, *LOC440356*, *SEZ6L2*, *ASPHD1*, *KCTD13*, *TMEM219*, *TAOK2*, *HIRIP3*, *INO80E*, *DOC2A*, *C16orf92*, *FAM57B*, *ALDOA*, *PPP4C*, *TBX6*, *YPEL3*, *GDPD3*, *MAPK3*, *LOC100271831*, *CORO1A*,*LOC613037*, *LOC595101*, *CD2BP2*, *TBC1D10B*, *MYLPF*, *SEPT1*, *ZNF48*, *ZNF771*, *DCTPP1*, *SEPHS2*, *ITGAL*, *ZNF768*, *ZNF747*, *ZNF764*, *ZNF688*, *ZNF785*, *ZNF689*, *PRR14*, *FBRS*, *SRCAP*, *SNORA30*, *PHKG2*, *C16orf93*, *RNF40*, *ZNF629*, *BCL7C*, *MIR762*, *CTF1*, *NCRNA00095*, *FBXL19*, *ORAI3*, *SETD1A*, *HSD3B7*, *STX1B*, *STX4*, *ZNF668*, *ZNF646*, *POL3S*, *VKORC1*, *BCKDK*, *MYST1*, *PRSS8*, *PRSS36*, *FUS*, *PYCARD*, *TRIM72*, *PYDC1*, *ITGAM*, *ITGAX*, *ITGAD*, *COX6A2*, *ZNF843*, *ARMC5*, *TGFB1I1*, *SLC5A2*, *C16orf58*, *AHSP*, *CSDAP1*, *C16orf67*	P
AUT71	Asperger, left sensorineural hearing loss, astigmatism, obesity, and speech abnormalities	arr[hg19] 15q11.2(22880274_23648846)x3	768.5	*CYFIP1*, *NIPA2*, *NIPA1*, *WHAMML1*, *GOLGA8IP*, *HERC2P2*, *HERC2P7*, *GOLGA8E*	VUS-LP
AUT81	ASD, psychomotor delay, and microcephaly	arr[hg19] 15q26.1(90719940_93509788)x1	2780	*SEMA4A, CIB1*, *GDPGP1, NGR1, IQGAP1, CRTC3, FURIN, RECQL1, FES, MAN2A2, RCCD1,PEC1, BLM*, *UNC45A, VPS33B*, *CHD2, SV2B, SLCO3A1, STX, RGMA*	P
AUT90	ASD, Mild ID, expressive dysphasia	arr[hg19] 22q13.33(51123491_51178264)x1	54.77	* SHANK3 *	P
AUT95	ASD, ID	arr[hg19] Xp11.22(52841226_5308135)x4	240.1	*XAGE5*, *XAGE3*, *FAM156A*, *FAM156B*, *GPR173.*	P
		arr[hg19] 9q34.13(134378819_134399206)x3	20.3	*POMT1*, *UCK1*	LP
		arr[hg19] 22q11.23(24276174_24332956)x3	56.7	*GSTT2*, *GSTT2B*, *DDTL*, *DDT*.	B
AUT97	ASD	arr[hg19] 15q11.2-q13.1(23641200_29331964)x3	5690	*MKRN3*, *MAGEL2*, *NDN*, *SNRPN*, *UBE3A*, *GABRB3*, *GABRA5*, *OCA2*, *HERC2, APBA2, NSMCE3*	P
AUT116	ASD	arr[hg19] 2q11.1-q13(95635960_110301275)x2-3 mosaic 75%	14,660	*ASTL, STARD7, TMEM127*, *SNRNP200*, *NCAPH*, *LMAN2L, CNNM4*, *ZAP70*, *VWA3B*, *CNGA3*, *C2orf64*, *TSGA10*, *LIPT1*, *AFF3*, *POU3F3*, *SLC5A7*, *RANBP2*, *EDAR, NPHP1*	P
AUT136	ASD (one brother with ASD, another with ID, mother with schizophrenia)	arr[hg19] 15q11.2-q13.1(23641200_29331964)x3	5690	*MKRN3*, *MAGEL2*, *NDN*, *SNRPN*, *UBE3A*, *GABRB3*, *GABRA5*, *OCA2*, *HERC2, APBA2, NSMCE3*	P
AUT167	ASD, short stature, ID, and macrocephaly	arr[hg19] Xp22.33-p22.2(60679_9711656)x0	9600	*PLCXD1*, *GTPBP6*, *NCRNA00107*, *PPP2R3B*, *SHOX*, *CRLF2*, *CSF2RA*, *IL3RA*, *SLC25A6*, *NCRNA00105*, *ASMTL*, *P2RY8*, *SFRS17A*, *ASMT*, *DHRSX*, *ZBED1*, *CD99*, *XG*, *XGPY2*, *XG*, *GYG2*, *ARSD*, *ARSE*, *ARSH*, *ARSF*, *MXRA5*, *PRKX*, *NLGN4X*, *VCX3A*, *HDHD1A*, *STS*, *VCX*, *PNPLA4*, *MIR651*, *VCX2*, *VCX3B*, *KAL1*, *FAM9A*, *FAM9B*, *TBL1X*, *GPR143, SHROOM2, CLCN4.*	P
AUT169	ASD, severe ID, absence of speech, macrocephaly, and joint laxity	arr[hg19] 3p25.3-p25.2(10975732_11885558)x1	910	*SLC6A11*, *SLC6A1*, *HRH1*, *ATG7*, *VGLL4*, *C3orf31.*	LP
AUT177	ASD, ID, and psychomotor delay	arr[hg19] 16p11.2(29350787_30198123)x3 90% mosaic dn	850	*BOLA2, SLX1B, SLX1A, SPN, QPR1, ZG16, KIF22, MAZ, PRRT2, PAGR1, MVP, CDIPT, SEZ6L2, KCTD13, TAOK2, HRIP3, DOC2A, C16orf92, TLCD38, ALDOA, PPP4C, TBX6,YPEL3, GDPD3, MAPK3, CORO1A, SULT1A4, SULT1A3*	P
AUT185	ASD	arr [hg19] 7q21.13(89865734_90897065)x3	1030	*STEAP2*, *C7orf63*, *GTPBP10*, *CLDN12*, *CDK14*, *FZD1.*	VUS
		arr[hg19] 22q11.21 (19710115_19755528)x3	45	*SEPT5*, *GP1BB*, *TBX1*	LP
AUT194	ASD and absence of speech	arr[hg19] 9p24.3(220253_315028)x3	94.7	* DOCK8 *	LP
AUT218	ASD, psychomotor delay	arr[hg19] 2q23.1(148790680_148879681)x1	90	* MBD5 *	LP
AUT219	ASD	arr[hg19] 8q21(82572090_82618539)x1	46.45	*IMPA1*, *SLC10A5*, *ZFAND1*	LP
		arr[hg19] 10q21.3(68516519_68613775)x1	97.26	* CTNNA3 *	LP
AUT223	ASD and mild ID	arr[hg19] 22q13.33(51116128_51219009)x1	102.88	* SHANK3 *	P
AUT228	ASD, hypotonia, and psychomotor delay	arr[hg19] 15q13.3(32098670_32423858)x1	325.18	*CHRNA7*	LP
AUT230	ASD and psychomotor delay	arr[hg19]15q11.2(22,880,274_23,648,846)x3	768.5	*CYFIP1*, *NIPA2*, *NIPA1*	VUS-LP
		arr[hg19] 15q13.3(32011762_32514890)x3	503	*CHRNA7*	VUS
		arr[hg19] Xq22.2(103220412_103269195)x2	49	*MIR1256*, *TMSB15B*, *H2BFXP*, *H2BFWT*	LB
AUT232	ASD, hypotonia, psychomotor delay, and myoclonus	arr[hg19] 3q29(195697011_197335597)x1	1610	*TNK2*, *TPRC*, *ZDHH19*, *SLC51A*, *TM45F19*, *UBXN7*, *SMCO1*, *WDR53*, *FBX045*, *CEP19*, *PIGX*, *PAK2*, *NCBP2*, *PIGZ*, *MELTF*, *DLG1*, *MIR4797*, *RUBCN*, *BDH1*, *KIAA8226*, *FYTTD1*, *BDH1*	LP

B—benign; LB—likely benign; VUS—variant of uncertain significance; P—pathogenic; LP—likely pathogenic. OMIM genes are underlined.

**Table 2 genes-14-00820-t002:** Detailed description of the VUS CNVs detected in the cohort.

Patient	Results	Size (Kb)	Genes. OMIM Genes Underlined
AUT7	arr[hg19] 22q11.21(18889039_19010508)x3	121	*DGCR6*, *PRODH*, *DGCR5*, *DGCR9*, *DGCR10*
AUT29	arr[hg19] 6q11.1 (61982931_62917272)x3	934.3	*KHDRBS2*
AUT43	arr[hg19] 5q21.2-q21.3(103185551_107582929)x1 mat	4390	*RAB9BP9*, *EFNA5, FBXL17*
	arr[hg19] Xq22(107976048_107979726)x2 dn	3.68	*IRS4* (benign)
AUT55	arr [hg19] 14q31.1 (80521415_81855721)x3	1334	*DIO2*, *C14orf145*, *TSHR*, *GTF2A1*, *SNORA79*, *STON2*
AUT56	arr [hg19] 4p16.3 273646_1033610)x3	759.96	*ZNF732*, *ZNF141*, *ABCA11P*, *ZNF721*, *PIGG*, *PDE6B*, *ATP5I*, *MYL5*, *MFSD7*, *PCGF3*, *CPLX1*, *GAK*, *TMEM175*, *DGKQ*, *SLC26A1*, *IDUA*, *FGFRL1*
	arr[hg19] 1p36.33(1627899_1663760)x1	35.8	*CDK11B*, *MMP23A*, *CDK11A* and *SLC35E2* (benign)
AUT62	arr[hg19] 14q13.2 (35404289_35780723)x3pat	376.4	*C14orf19*, *SRP54*, *FAM177A1*, *PPP2R3C*, *KIAA0391*, *PSMA6*
AUT75	arr[hg19] 1q44(249,104,658_249,212,668)x4	108	*SH3BP5L*, *MIR3124*, *ZNF672*, *ZNF692*, *PGBD2*
	arr[hg19] 5p14.3(21,269,398_21,691,673)x3 mat	422.77	*GUSBP1* (benign)
	arr[hg19] 17p12(15,183,668_15,185,073)x1, mat	1.40	(benign)
	arr[hg19] Xq22.2(103,220,412_103,269,195)x3, mat	48.78	*MIR1256*, *TMSB15B*, *H2BFXP*, *H2BFWT* (benign)
AUT83	arr [hg19] 2q11.2 (97728447_98021593)x1,	295	*FAHD2B*, *ANKRD36* (benign)
	arr [hg19] 4q25 (110170903_110390360)x3	219	*COL25A1*, *SEC24B*, (benign)
	arr [hg19] 10p12.1 (27613431_27694710)x1	81	*PTCHD3* (benign)
	arr[hg19] 17p13.3(1693_296541)x3	293	*DOC2B*, *RPH3AL*, *C17orf97*, *FAM101B*
AUT98	arr[hg19] 20p13(4666856_4674849)x1	7.99	* PRNP *
AUT100	arr[hg19] 1p32.3(54231745_54325010)x3	93.26	*TMEM48*, *YIPF1*
AUT103	arr[hg19] 8p11.21.1(429374004_43055737)x1	118.33	*FNTA*, *SGK196 (POMK)*, *HGSNAT*
AUT104	arr[hg19] 8p11.21.1(429374004_43055737)x1	118.33	*FNTA*, *SGK196 (POMK)*, *HGSNAT*
AUT129	arr[hg19] 10q24.32(103366911_103453040)x3	86.13	*DPCD*, *FBXW4.*
AUT146	arr[hg19] 15q12(27311913_27340486)x3	28.6	* GABRG3 *
AUT148	arr[hg19] 14q24.3(74036805_74550384)x3	513.58	*ACOT2*, *ACOT4*, *ACOT6*, *DNAL1*, *PNMA1*, *C14orf43*, *PTGR2*, *ZNF410*, *FAM161B*, *COQ6*, *ENTPD5*, *C14orf45*, *ALDH6A1*
AUT152	arr[hg19] 12p13.33(323258_658266)x2-3 mat	335	*SLC6A12*, *SLC6A13*, *KDM5A*, *CCDC77*, *B4GALNT3*
AUT154	arr[hg19] 21q11.2(15588480_15624577)x3	36.1	*RBM11*
AUT175	arr[hg19] 17p11.2-p11.1(21193939_22205821)x3, pat	1010	*MAP2K3*, *KCNJ12*, *KCNJ18*, *C17orf51*, *FAM27L*, *FLJ36000*
AUT179	arr[hg19] 12q24.33(132321161_132811002)x3	489.8	*MMP17*, *ULK1*, *PUS1*, *EP400*, *SNORA49*, *EP400NL*, *DDX51*, *NOC4L*, *GALNT9*
AUT180	arr[hg19] 12q24.33(132321161_132811002)x3	489.8	*MMP17*, *ULK1*, *PUS1*, *EP400*, *SNORA49*, *EP400NL*, *DDX51*, *NOC4L*, *GALNT9*
AUT195	arr[hg19] Xp22.2(13567942_13767555)x3	200	*EGFL6*, *RAB9A*, *TRAPPC2*, *OFD1*
	arr[hg19] Xq21.33(93842754_94388922)x1	546.17	-
AUT224	arr[hg19] 15q13.3(32019696_32575866)x3	556.17	*CHRNA7*
AUT225	arr[hg19] 15q13.3(32011762_32575866)x3	564.10	*CHRNA7*
	arr[hg19] 16p12.2(22629047_22709775)x3	80.73	*-*
	arr[hg19] 15q13.2(30938215_30971589)x3	33.37	*-*
AUT227	arr[hg19] 15q13.3(32011762_32514890)x3	503.00	*CHRNA7*
AUT229	arr[hg19] 15q13.3(32059475_3253966)x3	480.20	*CHRNA7*
AUT231	arr[hg19] 15q13.3(32011762_32514890)x3	503.22	*OTUD7A*, *CHRNA7*

**Table 3 genes-14-00820-t003:** Total CNVs detected in the analyzed cohort. CNVs are divided into deletions and duplications and shown with relative median size.

**CNV Type**	**DELETIONS**	**#CNV ˂ 100 Kb/Median SIZE (Kb)**	**#CNV 100–1000 Kb/Median SIZE (Kb)**	**#CNV ˃ 1000 Kb/Median SIZE (Kb)**
B/LB	32	22/28.74	8/159.62	2/1290
VUS	2	1/8.00	0/-	1/4390
P/LP	16	4/58.42 *	7/400.61 *	5/4643.80 *
Total	50	27	15	8
**CNV Type**	**DUPLICATIONS**	**#CNV˂100 Kb/Median SIZE (Kb)**	**#CNV 100–1000 Kb/Median SIZE (Kb)**	**#CNV ˃ 1000 Kb/Median SIZE (Kb)**
B/LB	37	21/48.90	15/224.64	1/1024
VUS	25	6/50.0	16/460.93	3/1124.67
P/LP	18	3/53.33	7/669.10 *	8/5630.1 *
Total	80	30	38	12

CNV—copy number variation; B—benign; LB—likely benign; VUS—variant of uncertain significance; P—pathogenic; LP—likely pathogenic. * indicates statistically significant values, *p* < 0.05.

## Data Availability

Not applicable.

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
