# Peer review of "Chromosomal Microarray in Patients with Non-Syndromic Autism Spectrum Disorders in the Clinical Routine of a Tertiary Hospital"

_genes, 2023, doi:10.3390/genes14040820_

Round 1

Reviewer 1 Report

1.      In abstract section, line 13-34, it is encouraged to only consist of 1 paragraph.

2.      Line 18 in abstract section, please do not use “we”. Make it into passive for more scientifically sound.

3.      Please end your abstract with a "take-home" message.

4.      Rearrange keywords alphabetically.

5.      Abbreviation as a keyword is not recommended and encouraged to be changed become a stand for its abbreviation.

6.      It is unclear whether the author's something new in this work. According to evaluation, several published studies by other researchers in the past adequately explain the issues you made in the present paper. Please be careful to highlight in the introduction section anything really innovative in this work.

7.      Previous study related needs to explain in the introduction section consisting of their work, their novelty, and their limitations to show the research gaps that intend to be filled in the present study.

8.      Line 70, for “we”, make it into passive.

9.   Develop the end of introduction section in line 70-71.

10.   To help the reader grasp the study's workflow more easily, the authors could include more visuals to the materials and methods section in the form of figures rather than sticking with the text that now predominates.

11.   What is the baseline of patient selection? Is there any protocol, standard, or basis that has been followed? It is unclear since the patient is very heterogeneous with a small number. The resonance involved impacts the present result makes this study flaws. One major reason for rejecting this paper.

12.   It is required to include additional information on tools, such as the manufacturer, the country, and the specification.

13.   The revised manuscript after peer review must provide detailed information on the error and tolerance of the experimental equipment utilized in this study. Due to the disparate outcomes of other researchers' subsequent studies, it would make for a valuable discussion.

Author Response

Response to Reviewer 1 Comments

Point 1. Extensive editing of English language and style required

Answer: I would like to thank the Rev1 for the comments trying to improbe the quality of the manuscript.i In this sense, an extensive English languaje and style review has been made.

Point 2. Comments and Suggestions for Authors

  1. In abstract section, line 13-34, it is encouraged to only consist of 1 paragraph.

Response: The abstract has been modified acording to the requirement.

Abstract: Autism spectrum disorders (ASD) comprise a group of neurodevelopmental disorders (NDD) characterized by deficits in communication and social interaction, as well as repetitive and restrictive behaviors. The genetic implications of ASD have been widely documented and numerous genes have been associated with it. The use of chromosomal microarray analysis (CMA) has proven to be a rapid and effective method for detecting both small and large deletions and duplications associated with ASD. In this article, we present the implementation of CMA as a first-tier test in our clinical laboratory for patients with primary ASD over a prospective period of four years. The cohort was composed of 212 individuals over 3 years of age, who met DSM-5 diagnostic criteria for ASD. The use of a customized array-CGH design (KaryoArray®), found 99 individuals (45.20%) with copy number variants (CNVs); thirty-four of them carried deletions (34.34%) and 65 duplications (65.65%). Twenty-eight of 212 patients had pathogenic or likely pathogenic CNVs, representing approximately 13% of the cohort. In turn, 28 out of 212 (approximately 12%) had variants of uncertain clinical significance (VUS). Our findings involve clinically significant CNVs, known to cause ASD (syndromic and non-syndromic), and other CNVs previously related to other comorbidities such as epilepsy or intellectual disability (ID). Lastly, we observed new rearrangements that will enhance the information available and collection of genes associated with this disorder. Our data also highlight that CMA could be very useful in diagnosing patients with essential/primary autism, and demonstrate the existence of substantial genetic and clinical heterogeneity in non-syndromic ASD individuals, underscoring the continued challenge for genetic laboratories in terms of molecular diagnosis.

  1. Line 18 in abstract section, please do not use “we”. Make it into passive for more scientifically sound.

Response: done.

  1. Please end your abstract with a "take-home" message.

Response: done a take-home message has been included.

………..Our data also highlight that CMA could be very useful in diagnosing patients with essential/primary autism, and demonstrate the existence of substantial genetic and clinical heterogeneity in non-syndromic ASD individuals, underscoring the continued challenge for genetic laboratories in terms of molecular diagnosis.

  1. Rearrange keywords alphabetically.

Response: Keywors has been alphabetically rearranged

Keywords: autistic spectrum disorder, copy number variations, microarray, tertiary hospital

  1. Abbreviation as a keyword is not recommended and encouraged to be changed become a stand for its abbreviation.

Response: we avoided the abbreviations. Keywords: autistic spectrum disorder, copy number variations, microarray, tertiary hospital

  1. It is unclear whether the author's something new in this work. According to evaluation, several published studies by other researchers in the past adequately explain the issues you made in the present paper. Please be careful to highlight in the introduction section anything really innovative in this work.

Response: We have changed the introduction addressing the changes suggested by the reviewer the novelty of this study was in the management of the primary/nonsyndromic ASD individuals

  1. Previous study related needs to explain in the introduction section consisting of their work, their novelty, and their limitations to show the research gaps that intend to be filled in the present study.

Response: We have changed the introduction addressing the changes suggested by the reviewer the novelty of this study was in the management of the primary/nonsyndromic ASD individulas. We present ed previous studies in spain and other countries (in the discussion section in the previous version)

  1. Line 70, for “we”, make it into passive.

Response: done

In this study we aim the use of CMA as a first-tier test for patients with primary ASD in a tertiary hospital, in a prospective

  1. Develop the end of introduction section in line 70-71.

Response: done.

In this study we aim the use of CMA as a first-tier test for patients with primary ASD in a tertiary hospital, in a prospective period of four years, in order to assess putative genes or regions involved, and evaluating the efficiency of CMA as a first-tier test for individuals with non-syndromic ASD.

  1. To help the reader grasp the study's workflow more easily, the authors could include more visuals to the materials and methods section in the form of figures rather than sticking with the text that now predominates.

Response: done. A new figure (Figure 1) was developed into the material and methods as an algorithm to facilitate manuscript comprehension.

  1. What is the baseline of patient selection? Is there any protocol, standard, or basis that has been followed? It is unclear since the patient is very heterogeneous with a small number. The resonance involved impacts the present result makes this study flaws. One major reason for rejecting this paper.

Response: done. The paragraph in Material and Methods (Subjects) has been revised and changed and a new figure (Figure 1) was developed into the material and methods as an algorithm to facilitate manuscript comprehension.

  1. It is required to include additional information on tools, such as the manufacturer, the country, and the specification.

Response: done.

  1. The revised manuscript after peer review must provide detailed information on the error and tolerance of the experimental equipment utilized in this study. Due to the disparate outcomes of other researchers' subsequent studies, it would make for a valuable discussion.

Response: done. A revised discussion has been included. And a limitation section has been included in Materials and Methods.

Reviewer 2 Report

The authors present data from chromosomal microarray analysis of a series of patients aged >= 3 with a clinical diagnosis of an autism spectrum disorder.  Findings resulted in a definitive genetic diagnosis in approximately 13 % of patients, consistent with previous published results.

There are a few minor discrepancies that need to be addressed in the manuscript.

1. The numbers in the text, page 4 lines 154--156, do not agree exactly with the numbers shown in Fig. 2.  for example, the text states that 12.7% of patients had P/LP CNVs while Fig. 2 suggests that 13.2% of patients had such changes.  This discrepancy needs to be addressed.

2. The number of VUS identified seems somewhat low compared to other studies.  The authors should address the potential reason for this.  For example, does their use of a relatively low-density array compared to other commercially available arrays (CytoScan HD, Illumina arrays) decrease their ability to detect VUS that may in the future be reclassified as P/LF (or B/LB)?

Overall this is a well-written manuscript that provides useful information regarding a population on which there have been a limited number of publications.

Author Response

Response to Reviewer 2 Comments

Point 1. Extensive editing of English language and style required

Answer: I would like to thank the Rev2 for the comments trying to improbe the quality of the manuscript.i In this sense, an extensive English languaje and style review has been made.

Point 2. Comments and Suggestions for Authors

The authors present data from chromosomal microarray analysis of a series of patients aged >= 3 with a clinical diagnosis of an autism spectrum disorder.  Findings resulted in a definitive genetic diagnosis in approximately 13 % of patients, consistent with previous published results.

There are a few minor discrepancies that need to be addressed in the manuscript.

  1. The numbers in the text, page 4 lines 154--156, do not agree exactly with the numbers shown in Fig. 2.  For example, the text states that 12.7% of patients had P/LP CNVs while Fig. 2 suggests that 13.2% of patients had such changes.  This discrepancy needs to be addressed.

Response: done. We address the discrepancies as a consequence of mixing different versions of the article we apologize fro the inconvenience.

  1. The number of VUS identified seems somewhat low compared to other studies.  The authors should address the potential reason for this.  For example, does their use of a relatively low-density array compared to other commercially available arrays (CytoScan HD, Illumina arrays) decrease their ability to detect VUS that may in the future be reclassified as P/LF (or B/LB)?

Response: done. Regarding this aspect (VUS) and the platform a new paragraph has been included in the discussion section.

Overall this is a well-written manuscript that provides useful information regarding a population on which there have been a limited number of publications

We thank the reviewer for his/her comments.

Round 2

Reviewer 1 Report

Following comments is given in this stage as follows:

1.      The authors encouraged to brief discuss anxiety since it is having relation to autism spectrum disorder. Please refer the relevant research as follows. doi: 10.3390/bioengineering9040157 and 10.3390/bioengineering9020048

2.      Results comparison with similar previous studies needs to give.

3.      Overall, discussion in the present article is extremely poor. The Authors must extend their discussion and make a comprehensive explanation. Just not simply mention the results with brief explanation.

4.      Before moving on to the conclusion section, the present study's limitation must be added at end of the discussion section.

5.      In the conclusion section, further research must be discussed.

6.      The authors need to enrich the reference from five years back. MDPI reference is strongly recommended.

7.      The authors occasionally created paragraphs in the entire document that were just one or two phrases long, which made the explanation difficult to understand. To make their explanation into a longer, more thorough paragraph, the authors should expand it. It is advised to use at least three sentences in a paragraph, with one serving as the primary sentence and the others as supporting phrases.

8.      The manuscript needs to be proofread by the authors since it has grammatical and language issues.

9.      Provide graphical abstract for submission after revision.
